# Improving the Evidence-Based Practice Skills of Entry-Level Physiotherapy Students through Educational Interventions: A Scoping Review of Literature

**DOI:** 10.3390/ijerph20166605

**Published:** 2023-08-18

**Authors:** Arben Boshnjaku, Solveig A. Arnadottir, Adrien Pallot, Marlies Wagener, Marja Äijö

**Affiliations:** 1Physiotherapy Department, University “Fehmi Agani” in Gjakova, 50000 Gjakova, Kosovo; arben.boshnjaku@uni-gjk.org; 2Department of Physical Therapy, University of Iceland, 102 Reykjavik, Iceland; saa@hi.is; 3Physiotherapy Department, Centre Européen d’Enseignement en Rééducation et Réadaptation Fonctionnelle, 93200 Saint-Denis, France; pallot.adrien@gmail.com; 4Institut d’Ingénierie de la Santé, Université de Picardie Jules Verne, 80000 Amiens, France; 5Center of Expertise Innovations in Care, Rotterdam University of Applied Science, 3015 EK Rotterdam, The Netherlands; m.n.wagener@hr.nl; 6School of Health Care, Savonia University of Applied Sciences, 70210 Kuopio, Finland

**Keywords:** physical therapy, entry level, physiotherapy education, evidence-based medicine, rehabilitation

## Abstract

Evidence-based practice (EBP) is an essential approach in healthcare, attracting growing interest among both practitioners and researchers. This scoping review aims to (1) systematically investigate the effectiveness of pedagogical methods used to facilitate learning of the EBP approach, and (2) explore the perceptions, experiences, and issues related to these learning methods. The overarching purpose is to identify the state of the art in pedagogical methods, instruments, influences, and barriers in teaching and learning EBP within entry-level physiotherapy education programs. This scoping review was conducted following PRISMA guidelines, with PubMed and Eric databases being searched for peer-reviewed original research articles using a combination of keywords. Excluding non-pertinent articles from the initial 465 identified, 12 were eligible for final inclusion (5 quantitative, 3 qualitative, and 4 mixed-methodology studies). A range of pedagogical methods and instruments for teaching EBP in physiotherapy education were detected, all of which having the capability to positively affect physiotherapy outcomes. Findings from this study support the significant influence that EBP exerts on the improving of the quality of teaching, together with the necessities that the involvement of EBP in physiotherapy education programs provide. Several barriers were identified, which should be taken into consideration when designing population-specific EBP strategies tailored to these particular needs.

## 1. Introduction

Evidence-based medicine (EBM) has been an emerging medical practice approach that has managed to continuously increase its influence. Its potential to converge towards all medical and allied health professions, and thus allowing them to benefit from EBM enhancements was initially mentioned in 1969 by Sackett [1]. Over the years, this convergence towards the new paradigm was realized. Nonetheless, with time going by, the term “medicine” became obsolete, mainly due to the wide variety of health professions using it, thus giving way to the term “evidence-based practice” (EBP).

Physiotherapy is one of the healthcare professions that aims to make people’s lives better and easier by promoting health and offering disease-targeting interventions. Physiotherapy education itself is a challenging and evolving process. In recent years, physiotherapy programs have incorporated EBP into their curricula. The aim is to enhance students’ clinical reasoning and decision-making skills, ultimately improving patient outcomes [2,3]. To date, there are many ways to teach students EBP reasoning [4]. However, no approach appears superior to another, and few articles are of good methodological quality [5]. Moreover, the possible effects of EBP learning, specifically for physiotherapy students, are little studied. Notably, McEvoy and colleagues [6] reported a gradual and significant increase in EBP knowledge from the first year to graduation within a 4-year entry level physiotherapy program, thus hinting at the potential positive effect education programs have on the matter.

Considering the importance of EBP within the health-related fields, its early integration within the teaching curricula of professional education presents the logical pathway to follow as a necessity to enhance the quality of education. Nonetheless, to date there seems to be a lack of information with respect to the teaching and learning of EBP in entry-level physiotherapy education programs. To enhance our understanding of current trends and the state of the art in this field, we conducted a scoping review with the two specific aims of (1) systematically investigating the effectiveness of pedagogical methods and (2) exploring physiotherapy students’ perceptions, experiences, and issues related to this mode of learning.

## 2. Materials and Methods

### 2.1. Protocol and Registration

The study was conducted by a team of experts representing different member institutions within the European Network of Physiotherapy in Higher Education (ENPHE) Research Group. To ensure a systematic methodology, the Joanna Briggs Institute Reviewers’ Manual—Methodology for JBI Scoping Reviews [7] was followed as a guideline. The Preferred Reporting Items for Systematic Reviews and Meta-Analyses extension for Scoping Reviews (PRISMA-ScR) checklist was used to guide the reporting of this review [8]. This review was not registered with PROSPERO. The definition of entry-level physiotherapy education included the cases involving students of level 6 from the European Qualification Framework (EQF) of the European Higher Education Area.

### 2.2. Eligibility Criteria

A list of specific eligibility criteria for articles to be included in the study were set a priori. Regarding the inclusion criteria, articles were included if

published in the English, French, German, or Italian language (as spoken by the authors) with full text;published in an internationally peer-reviewed journal that is indexed amongst the specified databases (PubMed and Eric);published within a timeframe between 11th of July 2012 and the 27th of March 2023;focused on educational interventions to increase EBP competence amongst physiotherapy students, as long as containing this set of two elements: entry-level physiotherapy students and EBP;quantitative and/or qualitative estimates of the effectiveness of the EBP educational approach on EBP-related competence, knowledge, attitudes, skills, and experiences were performed.

The studies not fulfilling the set criteria were excluded from/deemed not eligible to be involved in this scoping review.

### 2.3. Search Strategy

A search strategy protocol was developed a priori and unanimously agreed upon by all authors. The protocol was performed as follows:outlined key terms and Boolean operators to use;identified the electronic databases to conduct the search;identified the process for performing a systematic search;established the timeframe for the search;defined procedures for data extraction, processing, and interpretation.

University librarians from the respective institutions of the two experts that conducted the systematic search (University of Iceland and University “Fehmi Agani” in Gjakova) were consulted in order to avoid any confounders regarding the key terms and Boolean operators. The key terms were extracted from the descriptors of medical subject headings (MeSH), whereas Boolean operators (“AND”, “OR”, as well as additional instruments like “()” and “*”) helped conjugate the key terms with logical expressions. As a result, the final search equation used was as follows:

[(physical therapy OR physiotherapy OR rehabilitation) AND (education OR teach OR lear*) AND (EBP OR evidence-based practice) NOT (nurs*) NOT (midwife*)].

A systematic search was conducted simultaneously by two independent researchers (SAA and AB) within two major electronic databases: PubMed (https://pubmed.ncbi.nlm.nih.gov/ (accessed on 27 March 2023)) and Eric (https://eric.ed.gov/ (accessed on 27 March 2023)). The authors collectively chose these two specific databases based on their experience and perceived relevance for both physiotherapy teachers and students from the respective institutions. The process to perform the systematic search and conduct the selection of manuscripts was planned as follows:the articles identified through keywords from the databases were initially assessed for duplication, and the duplicates were removed;a screening based on titles and abstracts was conducted as an action to assess the eligibility criteria of the selected articles;a screening for eligibility criteria based on the full text of the remaining articles followed;the final list of remaining articles was included in the scoping review.

The conducted search encompassed a time period from the 11 July 2012 until the 27 March 2023. The search strategy aimed to identify papers from all types of research [7] where physiotherapy or physical therapy, education and evidence-based practice were aligned with each other. Outcomes were downloaded as individual Microsoft Excel files and compared between the two researchers, and then sent to follow the next steps for further analyses (as described above).

### 2.4. Study Selection and Data Extraction

Upon removing the duplicates, preliminary data analysis included a screening process based on titles and abstracts, followed by a full text reading upon selection. The processes of study selection and data extraction were conducted by two (other) independent reviewers (MA and AP) while following the above set criteria. A third reviewer (MW) was included in the process to solve ties in cases of disagreements. Final decisions for selection were taken with reviewers’ general consensus.

From the eligible studies, the following data were extracted:bibliographic information;study design;study population;educational intervention;duration of intervention;key findings and outcomes.

### 2.5. Synthesis of Results

We grouped the results of the articles according to the objectives of our review.

## 3. Results

### 3.1. Study Selection

The initial identification process using the set key terms and Boolean operators detected 397 and 69 articles within the PubMed and Eric databases, respectively. Amongst these, 7 duplicate records emerged and were removed. The remaining 459 articles were screened based on their titles and abstracts for the relevance to the set criteria, with 411 being excluded. Only 48 articles met the set criteria for eligibility to undergo the next step, retrieving their full texts and being further investigated. The full text analysis further excluded 36 articles after carefully analyzing the content. Amongst those, 17 were excluded due to participants not being entry-level physiotherapy students, 13 articles for not being based on original research, and 6 for not focusing on the topic of our review. Finally, a total of 12 studies were included, out of which 5 were of quantitative methodology, 3 were of qualitative methodology, and 4 were of mixed methodologies. The detailed article selection process is described in Figure 1—the flow diagram of search, identification, and selection throughout the scoping review process.

### 3.2. Effectiveness of Learning Methods (First Objective)

Five quantitative studies were included in this scoping review (Table 1). The oldest (within our pre-set timeline) was a study from 2014 by the team lead by Olsen and colleagues [9]. The authors compared the self-reported EBP behavior, abilities, and barriers by final-year physiotherapy students for the whole study period throughout clinical placements, revealing significant correlations between EBP exposure level and the students’ ability to appraise the research evidence (r = 0.41, *p* < 0.001). The association between EBP exposure level and the EBP behavior shown by the physiotherapy students was identified in elements like asking and searching for and critically appraising research evidence. Moreover, the EBP element of critical appraisal was identified as a potential barrier in the EBP process [9]. Regarding the perceived barriers (in particular) of EBP among physiotherapy students, another study reported the lack of time, poor understanding of statistical analysis, and lack of research skills, formal training, and access to paid articles together with the poor ability to critically appraise articles and inadequate infrastructure facilities as the major barriers towards practicing EBP [10]. Nonetheless, the authors proposed that effective education, especially when strategically incorporating EBP into physiotherapy teaching curricula, could serve as the most potent tool to overcome these barriers and achieve positive outcomes [10].

A study from Schweikhard and colleagues [11] assessed the positive significant influence that online library instructional tutorials implemented within an EBP graduate physiotherapy course (in addition to another occupational therapy course) can exert on students’ searching skills and their ability to find higher evidence levels. In particular, it was shown that more students from the post-tutorial cohorts were able to employ the advanced searching skills of using MeSH/subject terms and limits to identify and target evidence for their clinical query, as well as to cite higher-level studies and use fewer databases in their search [11]. The decrease in the number of databases used for their searches is another indicator of the influence that well-designed EBP teaching can have, enabling practitioners to be more specific and direct in their particular searches. Along the same line were the findings from Arienti and colleagues [12], who assessed the effectiveness of a digital tool for learning EBP competencies named EBP laboratory S4BE (Students 4 Best Evidence). The tool was utilized by entry-level physiotherapy students at the start and end of their education, resulting in significant improvements upon completion (*p* < 0.001). Such improvements were also reported individually in all domains of the evidence-based practice questionnaire (EBPQ) for pre-/post-educational intervention, including relevance, terminology, confidence, sympathy, and practice. Noteworthy, the highest improvements were reached in the domains of terminology (54% to 65%) and in practice (41% to 55%) [12].

Another rather comprehensive study published by Kloda and colleagues [13] followed the same trend of emphasizing the influence of EBP on teaching methodology and outcomes, though from the perspective of a new, alternative clinical question framework in comparison to the Patient, Intervention, Comparison, Outcome (PICO) model for improving students’ search skills, search results, and self-efficacy. The study was conducted through a randomized controlled trial (RCT) which physiotherapy and occupational therapy students underwent, while being randomly allocated to either a control group, learning about the PICO question framework, or an experimental group, learning about the alternative framework. The time length of the instructions was the same, with 90 min of information literacy instructed by a librarian, with respect to the pathway to formulate clinical questions and search the literature. Interestingly, no differences could be observed in search performance or search skills (strategy and clinical question formulation) between the two groups, including recall (*p* = 0.167), precision (*p* = 0.243), F-measure (*p* = 0.163), clinical question points (*p* = 0.653), search strategy (*p* = 0.676), identified concept questions (*p* = 0.178), and self-efficacy scores at both pre-instruction (*p* = 0.941) and post-instruction (*p* = 0.772).

**Table 1 ijerph-20-06605-t001:** Characteristics of the included quantitative studies.

Author(s), Year, Country	Aims	Participants	Intervention (Exposure) and Duration	Assessments	Key Findings
Arienti et al., 2021, Italy [12]	To evaluate the effectiveness of evidence-based practice (EBP) using Students 4 Best Evidence (S4BE) as an educational tool. To teach EBP competencies to undergraduate physiotherapy (PT) students.	The sample included 121 students completing a bachelor’s degree in PT at an Italian university.	Using S4BE platform as digital problem-based learning (DPBL) for EBP. Included were 24 educational training hours using S4BE blog, carried out over a 5-month period, with a total of 6 sessions, each 4 h in length. The first 5 sessions completed in first 3 months—last session completed after 2 months of clinical training.	All students completed an evidence-based practice questionnaire (EBPQ2), a validated tool for the evaluation of EBP competencies at the beginning of the laboratory (T0) and at the end (T1).	The students showed a significant improvement in all domains (*p* < 0.001), except in the sympathy domain, where the percentage score decreased from 71% to 60%. The best improvements were reached in the domains of terminology (54% to 65%) and in practice (41% to 55%).
Nair et al., 2021, India [10]	To identify perceived barriers to EBP among PT students.	A total of 182 final-year students, 112 interns, and 135 postgraduate students.	No intervention.	A total of 429 PT students participated in a survey where they self-reported barriers to EBP. The survey was a 12-item self-reported measure broadly divided into three domains: training-related barriers, organizational barriers, and personal barriers.	Most participants reported insufficient time, poor understanding of statistical analysis, and lack of research skills, formal training, and access to paid articles, along with poor ability to critically appraise articles, and bad infrastructure facilities as barriers towards practicing EBP.
Olsen et al., 2014, Norway [9]	To compare self-reported EBP behavior, abilities, and barriers during clinical placements reported by five cohorts of final-year PT students with different EBP exposure across the 3-year bachelor program.	PT total (*n* = 180) A cross-sectional study was conducted among five cohorts (2006–2010) with third year PT students at a university college in Norway.	No intervention.	Questionnaire with 42 items related to EBP behavior, ability, and barriers. The questionnaire contained three sections and took approximately 15 min to complete. Section 1 addressed background data such as sex, age, and access to internet. Section 2 and Section 3 consisted of items related to use of EBP during clinical placement.	Association between EBP exposure and students’ self-reported EBP behavior, abilities, and barriers was low for most items in the questionnaire. Correlations found for 8 items, related to information need, question formulation, use of checklists, searching, and ability to search for and critically appraise research evidence (strongest between EBP exposure level and ability to critically appraise research evidence (r = 0.41, *p* < 0.001).
Schweikhard et al. 2018, USA [11]	To measure how online library instructional tutorials implemented into an evidence-based practice course have impacted the information literacy skills of occupational therapy (OT) and physiotherapy graduate students.	PT and OT graduate students, total *n* = 180 following the EBP course during the period from 2012–2016.	Throughout the course occurrence (8 weeks in total), specifically designed online tutorials were introduced to PT and OT students as supplementary information to certain lecture topics.	A rubric was created to measure information literacy (IL) skills demonstrated in the students’ Step 5 Papers. The rubric scored components of strategies used in students’ searches and the three studies cited as the best evidence available for answering their clinical questions.	The rubric assessment of student Step 5 Papers reveals a statistically significant increase in student scores in their use of search terms and MeSH/subject headings (*p* = 0.00005), their use of limits (*p* = 0.03), and their citation of higher-level studies (*p* = 0.03).
Kloda et al., 2020, Canada [13]	To determine if a new, alternative clinical question framework was equally or more effective than patient, intervention, comparison, outcome (PICO) for improving students’ search skills, search results, and self-efficacy.	Completed both training and assessment: Control *n* = 34 Experimental *n* = 30 75 OT students and 76 PT students. These students were in either their final year of study toward their undergraduate degrees before a direct-entry master’s or in their qualifying year preceding the master’s degree program.	Randomized controlled trial (RCT) Two groups of PT and OT students assigned (control and experimental groups) each receiving 90 min of information literacy instruction from a librarian about formulating clinical questions and searching the literature using MEDLINE. Control group received instructions including the PICO question framework. Experimental group received instructions regarding the alternative framework (problem, intervention, population, outcome measure, time, context, professional stakeholder, and patient or family stakeholder).	At the outset of each instruction session, participants completed a demographic questionnaire and a short instrument measuring their information literacy self-efficacy (i.e., pre-test). This 12-item instrument was based on a longer, validated info-literacy self-efficacy instrument. Three weeks after the experimental and control groups received their instruction sessions, a data-gathering session was held for all participants. In this session, participants used the instrument for measuring their information literacy self-efficacy (i.e., post-test).	No differences in search performance or search skills (strategy and clinical question formulation) between groups. No differences in recall (*p* = 0.167), precision (*p* = 0.243), F-measure (*p* = 0.163), clinical question points (*p =* 0.653), search strategy (*p* = 0.676), identified concept questions (*p* = 0.178), or self-efficacy scores at pre-instruction (*p* = 0.941) or post-instruction (*p* = 0.772). Post-instruction self-efficacy scores were significantly higher than pre-instruction scores (*t*(63) = 2.627, *p* = 0.001). Alternative framework—as effective as PICO in teaching OT and PT students.

Abbreviations: DPBL, digital problem-based learning; EBP, evidence-based practice; EBPQ, evidence-based practice questionnaire; IL, information literacy; OT, occupational therapy; PICO, patient intervention comparison outcome; PT, physiotherapy; RCT, randomized controlled trial; S4BE, Students 4 Best Evidence.

### 3.3. Experiences of Learning Methods (Second Objective)

The qualitative studies included in the scoping review are described in Table 2. Compared to the other two studies, McMahon at al. included only final-year physiotherapy students [14], Johnson et al. included third- and second-year social education and occupational therapy students [15], whereas Olsen et al. also included twelve clinical instructors and four visiting teachers in physiotherapy [16] in addition to the physiotherapy students of different study years. Data were collected by focus group interview [15,16] and structured group feedback sessions [14].

The interventions used to increase students’ skills and knowledge related to EBP physiotherapy were different in all three studies. Johnson and colleagues [15] introduced and encouraged students to use the “EBPsteps app” during their clinical placements. In this case, there were no further assessments or requirements related to the use of EBPsteps. The study conducted by Olsen and colleagues [16] described a somewhat stratified approach of EBP inclusion within the curriculum while being distributed throughout the academic years. In the first year of study, physiotherapy students were introduced to the concept of EBP. They were encouraged to apply EBPsteps during their clinical placements as a support mechanism for writing and searching courses and tutorials. This strategy was primarily aimed at fostering a positive attitude towards EBP. The second year of study was characterized by more advanced EBP instruction, specifically focusing on the critical appraisal of research articles, clinical guidelines, and lectures on searching for research evidence. In the third year of study, principles of EBP were incorporated into several activities, such as discussing problem-based clinical scenarios and writing patient reports based on real clinical scenarios from clinical placements. Finally, the study by McMahon and colleagues [14] aimed to enhance students’ skills and knowledge related to EBP physiotherapy. The study particularly focused on assessing the experiences of final-year students with EBP teaching during their study years In these three studies, the duration of the EBP intervention varied from a single clinical training period to the entire duration of the degree program [14,15,16].

The results of the Johnson et al. study [15] showed that the information needs, academic requirements, and encouragement from clinical instructors triggered the students to use EBPsteps. However, the lack of EBP knowledge, academic demand, and emphasis on EBP in clinical placement were barriers to using EBPsteps. Design issues mattered, as use of the app was, for example, motivated by the opportunity to practice EBP in one place and taking notes in a digital notebook. The use of the app was hindered by anticipation that the use of phones during clinical placements would be viewed negatively by others and by specific design features, such as unfamiliar icons. In their study conducted in a primary healthcare practice, McMahon and colleagues [14] described how students felt empowered and confident in independently searching for literature and information in areas they believed were not thoroughly covered in the curriculum. Nevertheless, Olsen et al. [16] brought up the students’ attempt to apply EBP in clinical practice but for certain reasons prioritizing practice experience over EBP. Furthermore, the students’ perception of the need for clinical practice role models in EBP was additionally emphasized.

**Table 2 ijerph-20-06605-t002:** Characteristics of the included qualitative studies.

Author(s), Year, Country	Aims	Participants	Intervention (Exposure) and Duration	Assessments	Key Findings
Johnson et al., 2021, Norway [15]	To explore health and social care students’ experiences of learning about evidence-based practice (EBP) using the mobile application EBPsteps during their clinical placements.	Undergraduate students: social education (SE) *n* = 132 occupational therapy (OT) *n* = 26 physiotherapy (PT) *n* = 66	EBPsteps app, during one training period, no length in weeks	Four focus groups were conducted with a convenience sample of students from three undergraduate degree programs: occupational therapy, physical therapy, and social education.	The EBPsteps app was a relevant tool for learning EBP. Students need more knowledge and encouragement from clinical instructors to use EBPsteps app. The use of aps should be mandatory in the curriculum.
McMahon et al., 2016, Ireland [14]	To explore, using structured group feedback sessions (SGFS), final-year PT students’ perceptions of ability to work on a primary health care team upon graduation.	PT (*n* = 68) from four higher education institutions in Ireland participated.	EBP teaching in the curriculum. No specific intervention using EBP for the students who had primary health care experience. A four-year Bachelor of Science physiotherapy degree program	Group opinion obtained using structured 90 min group feedback sessions.	Final-year students felt that EBP teaching in the curriculum supported their learning and increased their confidence in their ability to take up employment in primary health care.
Olsen et al., 2013, Norway [16]	To explore beliefs, experiences, and attitudes related to third-year students’ use of EBP in clinical physiotherapy education among students, clinical instructors, and visiting teachers.	PT (*n* = 16) clinical instructors *n* = 9 visiting teachers *n* = 4	One-year introduction to EBP, two-year advance teaching to EBP, and three-year obligatory studies Three-year bachelor program (180 ECTS-credits)	Six focus group interviews were conducted: three with sixteen students, two with nine clinical instructors, and one with four visiting teachers. In addition, one individual interview and one interview in a pair were conducted with clinical instructors.	In clinical practice, students attempt to apply EBP, but they saw themselves as novices. Students prioritize their practice experience over learning to use EBP in clinical patients’ situations. Students perceived a need for role models in EBP.

Abbreviations: EBP, evidence-based practice; ECTS, European Credit Transfer and Accumulation System; OT, occupational therapy; PT, physiotherapy; SE, social education; SGFS, structured group feedback sessions.

Table 3 presents four mixed-method studies included on this scoping review, all of which report EBP interventions (exposure) amongst students in entry-level physiotherapy bachelor programs. In one of the studies, participants were both surveyed and interviewed 2–7 years after graduation from their physiotherapy program [17], while participants in the three other studies were assessed during their studies [6,18,19]. The EBP interventions and outcomes differed; hence, the results and key findings varied, although quantitative and qualitative findings were generally congruent in all studies and indicated positive effects on students’ competencies in EBP. Rotor and colleagues [17] aimed to determine the influence of an EBP educational experience on knowledge, attitudes, and practice of EBP among physiotherapy graduates (2–7 years after graduation). Quantitative and qualitative findings were congruent and revealed that EBP activities in the physiotherapy bachelor-level program had a positive impact on the students who graduated with good knowledge, confidence, and positive attitudes for EBP. However, these activities were not sufficient to help graduates to overcome barriers that hinder EBP uptake in their clinical practice, after graduation. The focus group participants pointed out that the lack of autonomy among physiotherapists in the Philippines might be the main barrier for EBP use in their practice. The researchers suggested strategies for improving EBP education, such as finding ways to include real-world demands.

In the study from Lennon et al. [19], the main aim was to assess the influence of changing early EBP instructions from content-based learning (with physiotherapy students as audience participants) to interactive problem-based learning (PBL). This was made possible through two student cohorts, one receiving the content-based approach and the other receiving the PBL approach within a certain 5 ECTS (European Credit Transfer and Accumulation System) module. Quantitative and qualitative findings were congruent and revealed that the PBL approach was effective at promoting early EBP. Students identified with the interactive, collaborative, and experiential nature of PBL for EBP instruction. All mean Likert scores relating to subject understanding, relevance of assessments, achievement of learning outcomes, teaching, and overall module satisfaction improved compared to the PBL content-based approaches. Students’ comments post-PBL continued to identify EBP as a difficult concept, but comments on the teaching and assessment approach were positive and focused on the collaborative nature of PBL, identifying EBP, communication, and team-working skills acquired, praising the real-life, practical application of EBP taken, and commenting on the improvement in EBP self-efficacy. The quantitative self-assessment results following a PBL further supported improvement in EBP domains of knowledge, confidence, and practice. The researchers concluded that a PBL is a feasible, collaborative approach for early EBP instruction in entry-level physiotherapy education.

Another study aiming to assess the potential changes in physiotherapy students’ EBP outcomes through a formal multifaceted EBP curriculum was conducted by McEvoy and colleagues [6]. The EBP training resulted in both qualitative and quantitative changes in participants’ knowledge and perceptions of EBP. The quantitative self-assessments showed improvements in EBP relevance, confidence, practice, and knowledge, and the test supported an improvement in measured EBP knowledge. The qualitative and quantitative findings were well aligned except for knowledge in statistical terminology, where the students’ self-assessed and tested understanding was better than reported by the focus group. The qualitative findings highlighted the importance of providing relevant clinical context and positive role models (lecturers and clinical educators) for students during EBP training. Four main descriptive themes were constructed: (a) a shift in thinking over time; (b) the need for relevance and context; (c) learning by doing; and (d) getting the timing right for EBP training in entry-level physiotherapy education, while encouraging educators in physiotherapy to use these findings to support decisions of timing and content of EBP curricula in entry-level programs.

The last study aiming to assess the potential changes in physiotherapy students’ EBP outcomes within the mixed-methodology group was the one conducted by Bozzolan and colleagues [18]. The intervention involved students from all 3 years of the entry-level bachelor program, with each group subjected to different combinations of three intervention components. The 1st-year students had formal EBP courses followed by internship EBP assignments. The 2nd-year students had formal EBP courses followed by internship EBP assignments and participated in an EBP journal club. The 3rd-year students received internship EBP assignments and participated in an EBP journal club. Both 1st- and 2nd-year students improved their knowledge and skills over the 6-month study period, while the 3rd-year students (no formal EBP courses) did not improve on the quantitative assessments. The integration of the EBP into student practice during an internship seemed to be hindered by the absence of direct examples by the clinical educators and the lack or delay of feedback. Nonetheless, internship documentation gave evidence of EBP clinical behaviors, while focus group interviews revealed that students valued EBP but perceived barriers, and clinical teachers were described as both barriers and facilitators, indicating that students should be assigned to clinical professionals able or at least willing to use EBP.

**Table 3 ijerph-20-06605-t003:** Characteristics of the included mixed-method studies.

Author(s), Year, Country	Aims	Participants	Intervention (Exposure) and Duration	Assessments	Key Findings
Rotor et al., 2020, Philippines [17]	To determine influence of evidence-based practice (EBP) on knowledge, attitudes, and practice among physiotherapy (PT) graduates. To compare graduates’ EBP clinical practice profile to their EBP knowledge and attitudes. To explore graduates’ views on their own EBP education and how it influenced their clinical practice.	Five cohorts of PT graduates from a five-year entry-level BS program. Quantitative survey and qualitative open-ended questions (*n* = 77). Qualitative, focus groups (*n* = 8).	EBP activities embedded in the curriculum of the fourth and fifth year.	After graduation (2–7 years), participants were surveyed and interviewed, providing quantitative and qualitative data.	EBP activities: positive impact on students graduating with good knowledge, confidence, and positive EBP attitudes, not sufficient for graduates to overcome barriers hindering EBP uptake in their postgraduate clinical practice. Lack of physiotherapists’ autonomy in Philippines—the main barrier for EBP.
Lennon et al., 2019, Ireland [19]	To assess the influence of changes in early EBP instructional delivery from content-based learning to an interactive problem-based learning approach.	Two cohorts of PT students in a four-year entry-level Bachelor of Science program.	One EBP module in the second year, titled “Application of Physical Agents”. In the first cohort, the module was content-based; in the second cohort, it was problem-based.	Following both modules, students completed a standard university anonymized online survey providing both quantitative and qualitative data. Before and after the problem-based module, quantitative self-report evidence-based practice profile questionnaire (EBPPQ) data were collected.	Practice-based learning (PBL) approach: effective at promoting early EBP, improvements in EBP knowledge, confidence, and practice, preferred by students identifying with interactive, collaborative, and experiential nature of PBL, improved students’ understanding, achievement of outcomes, teaching, increased overall satisfaction.
McEvoy et al., 2018, Australia [6]	To investigate and integrate quantitative changes in EBP outcomes among PT students and qualitative students’ perceptions of factors impacting changes in selected EBP outcomes.	A single cohort of PT students in a four-year entry level BS-program. Quantitative pre-post (N = 56). Qualitative, focus groups (N = 21).	All participants completed three formal and multifaceted EBP courses in the first, second and fourth year.	Before and after EBP training, quantitative self-report data were collected on actual EBP knowledge. After EBP training, qualitative data were collected on students’ perceptions of EBP courses, current use of EBP, and role models during the PT program.	EBP training: improved EBP relevance, confidence, practice, improved knowledge of research evidence competencies (K-REC) actual knowledge.
Bozzolan et al., 2014, Italy [18]	To evaluate EBP knowledge and skills among PT students. To investigate EBP clinical behaviors of the students. Explore the students’ EBP perceptions and attitudes.	Three different cohorts of first- (*n* = 26), second- (*n* = 28), and third- (*n* = 19) year PT students, in a three-year entry-level Bachelor of Science program (*n* = 73).	An integrated educational curriculum on EBP (seven months) formed by three concurrent courses offered to three different groups: Formal EBP courses (first and second years). Internship EBP assignments (all three years) EBP journal club (second and third years).	Before and after EBP curriculum, quantitative data collection in all groups using A-Fresno Test for rehabilitation professionals with six additional questions from the A-Fresno test for PT. During and after EBP curriculum, qualitative data collected by inspecting internship documents for EBP clinical behaviors and with interviews on EBP perceptions and attitudes.	Combination of formal courses and internship assignments improved EBP knowledge/skills. EBP activities only during internship did not result in improvements. Students in focus groups valued EBP but perceived many barriers, described clinical teachers as obstacles or models, and highlighted the importance of being assigned to professionals willing or able to use EBP.

Abbreviations: EBP, evidence-based practice; EBPPQ, evidence-based practice profile questionnaire; K-REC, knowledge of research evidence competencies; PBL, practice-based learning; PICO, patient intervention comparison outcome; PT, physiotherapy.

### 3.4. Influence of EBP in Physiotherapy Education

Finally, Table 4 provides a qualitative summary of the direct influence of EBP on entry-level physiotherapy education (positive versus negative outcomes) across all included studies, along with the self-reported perceived barriers as described by the participants/students. As can be observed, the positive influence of EBP was reported in all studies with no cases of negative outcomes related to EBP.

## 4. Discussion

This study provides an overview of the current state of the art of teaching and learning EBP within entry-level physiotherapy education programs. Its importance lies in the novelty it brings to the effectiveness of different pedagogical approaches and the good practices and implications from others’ experiences, thus suggesting potential pathways to follow. Noteworthy findings from this study support the significant influence that EBP exerts on the improving of the quality of teaching and the gaining of professional skills in physiotherapy practice, together with the necessities that the involvement of EBP in physiotherapy education programs provide.

The introduction of EBP within the teaching curricula emerged as an effective means to enhance the quality of teaching and to improve the students’ acquisition of theoretical knowledge and practical skills. The integration of EBP into physiotherapy education programs offers students a unique opportunity to learn directly from the best experiences and thus significantly enhance the quality of professional theoretical knowledge and practical skills. Amongst the selected studies within this systematic scoping review, the different pedagogical methods to teach and apply EBP in physiotherapy education programs were observed to be effective in all cases. This is fundamentally important, especially since the EBP is nowadays recommended as a relevant and integral component of physiotherapy education guidelines [20] and professional practice policy [21]. Having this in mind, EBP emerges as a unique pathway that can offer the enhancement of the quality of entry-level physiotherapy education. Furthermore, it emerges as a promising tool for enabling future experts to continuously update and evolve their theoretical knowledge and practical and clinical skills independently from their formal education. Nonetheless, a general consensus on the exact forms and types of EBP needs to be further reached and established.

One interesting finding that was brought to our attention was the effectiveness of different instruments that were used as means of teaching EBP in physiotherapy education programs [11,12,13,15,22]. One of the most commonly applied framework instruments for asking and classifying clinical questions when teaching the health professions is the patient, intervention, comparison, and outcome (PICO) tool [22,23]. PICO’s applicability encompasses rehabilitation professions including physiotherapy. However, it is important to note that the effectiveness of EBP in physiotherapy teaching extends beyond the conventional frameworks (e.g., PICO) that are used to ask and classify clinical questions. A typical example of this was reported within a rather comprehensive randomized controlled trial (RCT) by Kloda and colleagues [13] assessing the effectiveness of a novel alternative clinical question framework for search skills, search results, and self-efficacy for rehabilitation professions within education settings. Interestingly, they reported it to be as effective as PICO when assessing students’ searching skills, while observing the capability of these librarian-led workshops that were using frameworks centered on question formulation to increase information literacy and self-efficacy [13]. Other instruments such as EBPsteps [15], S4BE [12], or even online library tutorials [11] were just as effective in achieving goals through transmitting knowledge and the skills to perform. Thus, having in mind the effectiveness and learning potential that EBP encapsulates, the fact that EBP can be taught using a variety of instruments is encouraging for all those engaged in physiotherapy education.

This scoping review identified several perceived barriers that were particularly observed within five of the selected studies. Amongst those, time emerged as the most common perceived barrier [9,10,18], followed by the lack of scientific-related knowledge and skills [9,10,16,18], ability to critically approach different issues [9,10], lack of autonomy from the medical model of practice [17,18], and lack of appropriate infrastructure [10]. One particular study conducted with third-year physiotherapy students who were expected to apply EBP in their clinical placements [16] identified a lack of EBP culture as perceived amongst them, together with the emphasized need for EBP role models throughout clinical placements. Consecutively, the time component is a concern of particular interest that might be overcome. However, to do so, it is imperative to follow a more careful and well-planned daily schedule, as well as to undergo a significant routine change. Olsen and colleagues [9] identified the lack of time to search and to apply the research evidence as the major perceived barrier by the entry-level students of physiotherapy programs. But is this age and social group not the one often characterized by time abuse through their online surfing, social media scrolling, and gaming [24,25]? Having this in mind, we believe that time as a self-perceived barrier by students of entry-level physiotherapy education programs emerges as a potential target towards which future developments should be oriented. In fact, in addition to time planning, the science-related knowledge and skills, critical thinking, personal and professional autonomy, as well as infrastructural improvements are all improvable targets to better EBP in education programs.

Even though performing the study to the best of our knowledge and capabilities, certain limitations prevailed. One potential limitation could be the limited number of papers addressing the main research issue of this scoping review, coupled with the inclusion of individual case studies and the results derived from them. Nonetheless, these limitations might also create an opportunity to invite researchers to further explore the field of EBP within entry-level physiotherapy education. As more articles are published, it will be necessary to assess the methodological quality of the included studies. Another potential limitation of this study could be the inclusion of only two electronic databases in the search process. Undoubtedly, databases like Journal Citation Reports (JCR), Scimago Journal Rank (JCR), or Physiotherapy Evidence Database (PEDro) could have contributed towards finding more targeted articles as specified in the search. However, the decision to use the selected databases (PubMed and Eric) was made by consensus based on the experiences of the authors who represent five different countries and physiotherapy institutions within ENPHE. These databases were considered to be the most applicable and more usable two databases in their practice.

## 5. Conclusions

The key findings from this scoping review include the following:EBP exerts a positive influence on entry-level physiotherapy education programs, regardless of how it is integrated into the curriculum;EBP enhances the quality of teaching and facilitates students’ gains in theoretical knowledge and practical skills;There exists a range of perceived barriers to the application of EBP, from intellectual aspects (such as a lack of critical thinking, scientific knowledge, and skills) to technical aspects (including equipment, prioritization, and time constraints).

EBP has an important role within entry-level physiotherapy education programs. It helps improve the quality of teaching, as well as enhancing the transfer of quality and quantity of knowledge and practical skills. A range of pedagogical methods and instruments for teaching EBP in physiotherapy education exists, all having the capability to positively affect professional outcomes.

Time constraints, lack of science-related knowledge and skills, inability to critically approach different issues, as well as the lack of autonomy from the medical model of practice emerge as the highest self-perceived barriers by physiotherapy students.

These findings call for immediate actions to include EBP when lacking, as well as enhancing the level of EBP in physiotherapy teaching curricula. EBP should have a significant role within entry-level physiotherapy education programs, notwithstanding their form (BSc, BA, or professional diploma) or length (either 3 years—180 ECTS credits or more). The well-planned and population-based frameworks tailored to students and end-users’ specific needs will help to further develop and facilitate decision-making processes and communication skills with all stakeholders.

## Figures and Tables

**Figure 1 ijerph-20-06605-f001:**
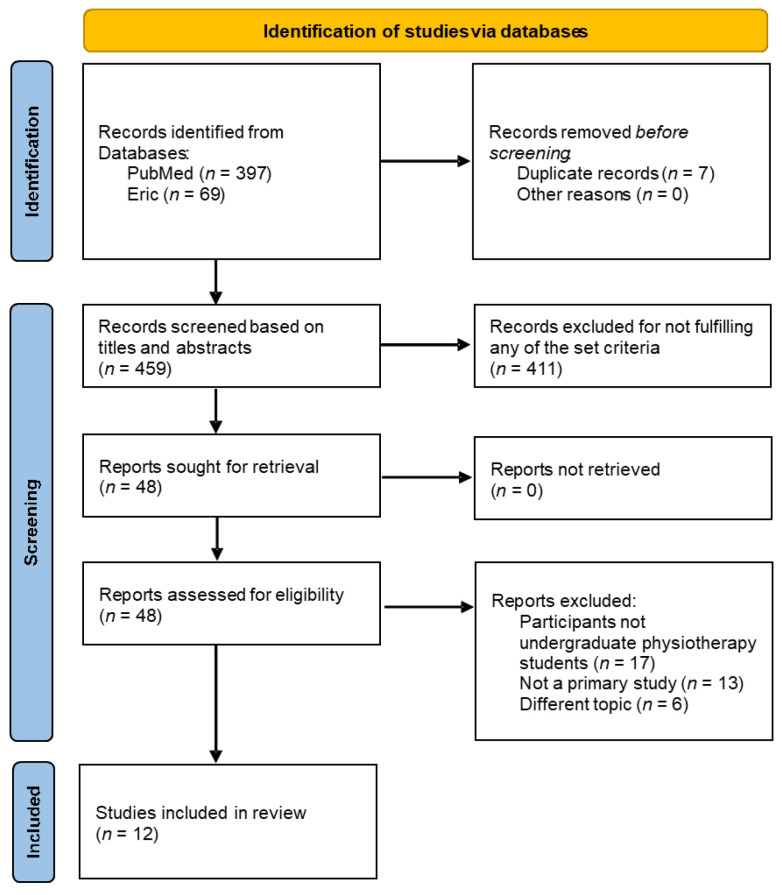
Article selection process following the PRISMA flow diagram.

**Table 4 ijerph-20-06605-t004:** The influence of evidence-based practice on physiotherapy educational outcomes, with an emphasis on potential barriers.

Evidence-Based Practice (EBP) Influence	EBP Outcome
Positive	9
Negative	0
Barriers	− lack of role models − novices in clinical practice − prioritizing practice experience over EBP* − access to equipment − critical thinking − knowledge of scientific approach − search skills and access to databases − time

Abbreviations: EBP, evidence-based practice.

## Data Availability

Not applicable.

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
