# Peer review of "Improving the Evidence-Based Practice Skills of Entry-Level Physiotherapy Students through Educational Interventions: A Scoping Review of Literature"

_ijerph, 2023, doi:10.3390/ijerph20166605_

Round 1
Reviewer 1 Report
1) I think it is advisable to better specify the objectives in the introduction section. It is not clear to me if there are 2 or 5.
2) There is talk of a search protocol initially established... and as a reader it has been difficult for me to know what that protocol is.
Author Response
Dear Reviewer 1
We would like to thank the reviewer for the very valuable comments and think that we have addressed all the issues that have been mentioned. In attachment is our response to yours detailed comments.

Reviewer 2 Report
Congratulations to the authors for a relevant and interesting work. I have some recommendations for you that I hope will help you.
The title needs to be more specific, providing an idea of what will be read. Including words like "literature review" or "systematic review" would help the reader understand what type of research is being conducted.
Why haven't databases like SJR and JCR been included?
It can be understood that the authors have tried to follow PRISMA for the review; however, they do not mention it nor follow all the steps indicated by this methodology. This is a significant problem for the potential replicability of the research.
In Figure 1, more information could be provided, such as the number of duplicates and the number excluded for each exclusion criterion considered, including text and abstract not being full-text, incorrect background, etc.
The results need to be rewritten; they are confusing to follow. It would be helpful to have a more structured presentation that aligns with the research objectives.
Why was the decision made to separate according to study design? This point has not been mentioned before. What is the objective?
In the discussion or conclusions, it would be good to provide key findings that assist potential readers in effectively applying evidence-based practice methodology.
I also believe that additional limitations need to be included. Some results are based on single case studies, and it is highly doubtful that findings from a single case study can be generalized.
I think a thorough revision of the manuscript is necessary before it can be published.
Author Response
Dear Reviewer 2
We would like to thank the reviewer for the very valuable comments and think that we have addressed all the issues that have been mentioned. In attachment is our response to yours detailed comments.

Reviewer 3 Report
Thank you for this review and for this work, as more and more academics talk about knowledge transfer to students.
Regarding the work, I think that a few small suggestions should be modified before publishing
Abstract
Re-structure the abstract based on the following aspects: cite the database, objectives, state results, do not cite the entire methodology, do not use acronyms
Methods
There is language bias and you should comment on it. Also comment why use only these two bases and not use a specific one like PEDro
Results
- The flowchart is not correct and also the data is not correct. The elimination process and the causes of the 36 excluded that have been read in full text must be described (prepare a table and add complementary material).
-Watch for misspellings in tables
- Quantitative results are very scarce. Also, to say nothing of Kloda et al., 2020, Canada [12]
- Again, qualitative results are very scarce.
Introduction, Discussion and Conclusion
Okay
I think that English is very understable
Author Response
Dear Reviewer 3
We would like to thank the reviewer for the very valuable comments and think that we have addressed all the issues that have been mentioned. In attachment is our response to yours detailed comments.

Round 2
Reviewer 2 Report
Congratulations to the authors for the work done. Taking into account that there have already been changes due to the contributions of other reviewers I only have specific contributions
I think it is necessary that it must be indicated that it is a review in the title
In the methodology it is necessary to include more details about the methods used in the search
What was the search equation used
The concrete search dates should be estacter at least month and hoop
What was the process to do the selection of manuscripts?
It has not been included.
Kind Regard
Author Response
Thank you for the good comments on our article. Please find attached our feedback to your comments.
